# How Does Agricultural Mechanization Service Affect Agricultural Green Transformation in China?

**DOI:** 10.3390/ijerph20021655

**Published:** 2023-01-16

**Authors:** Xuelan Li, Rui Guan

**Affiliations:** 1School of Economics and Management, Anhui Agricultural University, Hefei 230036, China; 2School of Management, Anhui Science and Technology University, Bengbu 233030, China; 3School of Politics and Public Administration, Zhengzhou University, Zhengzhou 450000, China

**Keywords:** agricultural mechanization, agricultural green transformation, smallholder, spatial spillover effect, China

## Abstract

Agricultural mechanization service (AMS) is a critical path to achieving agricultural green transformation with smallholders as the mainstay of agricultural production. Based on the panel data of 30 Chinese provinces from 2011 to 2020, this paper measures the AGTFP using the Super-SBM model and examines the effects of different AMS supply agents on AGTFP and spatial spillover effects through the spatial Durbin model. The main conclusions are as follows: First, China’s AGTFP showed a stable growth trend, with the mean value increasing from 0.1990 in 2011 to 0.5590 in 2020. Second, the specialization (SPO) and large-scale (LSO) of AMS supply organizations have significantly positive effect on the AGTFP of the local province. However, SPO has a significantly positive effect on the AGTFP of the neighboring provinces, while LSO has the opposite effect. Third, the specialization of AMS supply individuals (SPI) has significantly negative effect on the AGTFP of the local province. In contrast, the large-scale AMS supply individuals (LSI) has the opposite effect. Furthermore, the spatial spillover effects of both are insignificant. Fourth, the spatial spillover effect of AGTFP shows asymmetry among different regions and indicates that AMS resources flow from non-main grain production and economically developed regions to main grain production and less developed regions. These findings provide helpful policy references for constructing and improving the agricultural mechanization service system and realizing the agricultural green transformation in economies as the mainstay of agricultural production.

## 1. Introduction

Agricultural green transformation is a critical path for the continuous promotion of agricultural modernization and an essential foundation for human society to achieve sustainable development. Since the reform and opening up, China’s agricultural modernization process has made remarkable achievements. On the one hand, the productivity of Chinese agriculture has increased significantly [1]. According to statistics, China’s total grain production increased from 43.07 million tons in 2003 to 68.24 million tons in 2021, and China has achieved an increase in grain production for eighteen consecutive years. On the other hand, the industrial structure continues to be optimized. The value added of primary industry in China has decreased from 27.7% of GDP in 1978 to 7.3%, gradually approaching the average level of developed countries [2]. However, the rapid development of agricultural modernization in China has been accompanied by excessive use of agricultural chemicals, irrational utilization of agricultural waste, vast consumption of fossil energy, and soil destruction. These problems have led to severe agricultural non-point source pollution and carbon emission [3,4,5,6]. Meanwhile, as one of the most populous developing countries in the world, China uses 9% of the global arable land to feed nearly 21% of the global population [7]. The continued expansion of food demand further pushes the need for economies to achieve a green revolution centered on sustainable and intensive agricultural development [8,9,10].

The Chinese government attaches great importance to the agricultural green transformation. In the 13th Five-Year Plan, the word “green” was included in the national development concept for the first time. Subsequently, the Chinese government has repeatedly emphasized the importance of agricultural green transformation in several consecutive Central Government No. 1 documents and programmatic documents such as the 14th Five-Year Plan. It has built specific initiatives for agricultural green transformation to construct high-standard farmland, modern seed industry development, agricultural mechanization promotion, and action to reduce chemical fertilizers. However, the agricultural green transformation in China is complex, and the central problem lies in the pattern of smallholder production and operation formed by the household contract responsibility system. There are about 200 to 300 million smallholder farmers in China, and they have made significant contributions to China’s food security as the mainstay of agricultural production [11,12]. However, smallholders are closely associated with land fragmentation, highly dependent on high to excessive factor inputs, and inefficient production [13]. It has to be acknowledged that it is a relatively slow process to solve the problem of agricultural scale operations by land transfer. Currently, the development of the rural land transfer market in China still lags significantly behind the transfer of rural labor, and the proportion and total area of rural land transfer households urgently need improvement [14]. Based on this, socialized agricultural services with agricultural mechanization services (AMS) as the core are considered essential to solving the dilemma of “smallholder management” [15,16].

As early as 500 years ago, “wheat cutters” appeared in China that provided professional wheat-cutting services across hundreds of kilometers, the early agricultural social service in China. Moreover, with the rapid improvement of agricultural mechanization, the AMS system in China has also been rapidly developed. According to statistics, the total output value of the agriculture, forestry, animal husbandry, and fishery service industry snowballed from 287.34 billion yuan in 2011 to 702.98 billion yuan in 2020, of which the income from agricultural machinery operation services in 2020 was as high as 361.503 billion yuan, accounting for 51.42% of the total output value. AMS is essentially a specific form of vertical division of agriculture, which involves smallholders in modern agriculture and replaces land-scale operations with service-scale operations. Furthermore, AMS can break through the paradox of “smallholder management” in which small-scale decentralized operation characteristics cannot endogenize economies of scale [17]. Therefore, exploring the critical role of AMS in the agricultural green transformation can help increase smallholder productivity and reduce adverse environmental impacts to ensure food security. As a result, it can provide important policy implications for the agricultural green transformation in China and other countries with smallholders as the mainstay of agricultural production.

Most studies have visualized agricultural green transformation as agricultural green total factor productivity (AGTPF), and many valuable studies have been conducted in the following directions: (1) Measurement and decomposition of AGTFP. To measure AGTFP, we need to construct an indicator system from two dimensions: input and output. Input indicators mainly include labor, land, machinery, fertilizer, and agricultural energy consumption [18,19,20]. In the output dimension, compared with the traditional total factor productivity in agriculture, the measurement of AGTFP incorporates undesired outputs into the indicator system, mainly including carbon emissions and agricultural non-point source pollution [21]. However, some scholars argue that it is questionable whether livestock farming and organic solid waste should be considered as pollution in agricultural non-point source pollution [22]. On this basis, the decomposition of AGTFP reveals that technological progress is considered the driving force for the continuous growth of AGTFP in China [23], while declining technical efficiency and scale efficiency slow down the growth of AGTFP [24]. (2) Spatiotemporal characteristics of AGTFP. Existing studies have reached a certain consensus on the spatiotemporal evolution characteristics of AGTFP in China. Since the reform and opening up, China’s AGTFP has shown a fluctuating growth trend, but the gaps among regions have also gradually expanded [25,26,27]. AGTFP in eastern, central, and western China decreases in order [22,28,29]. (3) Analysis of the driving mechanism of AGTFP. Changes in factor inputs directly contribute to AGTFP, including farm size expansion, green manure application, agricultural mechanization increase, and human capital input [30,31,32,33]. At the same time, policy interventions are also essential to address factor input overload and environmental damage externalities. Existing studies have empirically tested that implementing crop insurance, carbon trading systems, and other agricultural environment protection policies can help increase AGTFP [18,19,34]. In addition, the use of new technologies is also a significant driving force for the sustained growth of AGTFP, including renewable energy consumption, the spread of Internet technologies, and the development of inclusive digital finance [35,36,37]. Some scholars have also studied the influencing factors that hinder the growth of AGTFP, including factor market distortion, resource mismatch, farmland pollution, agricultural disasters, and climate change [38,39,40,41].

In addition, some studies have also explored the critical role of agricultural services in promoting agricultural green transformation. Xu et al. characterized agricultural services in terms of agricultural services output and indicated that agricultural services helped to promote the growth of AGTFP, and the improvement effect was higher in the eastern region [42]. Zhu et al. characterized productive agricultural services in terms of the number of people employed in agricultural services and indicated that productive agricultural services effectively improved agri-environmental efficiency by optimizing inputs and increasing outputs [43]. From a micro perspective, Li et al. showed that adopting green agricultural services by farmers helped improve AGTFP, and the improvement effect was more substantial with higher adoption [44]. Zhang et al. showed that fertilizer application services and pest control services were crucial aspects of promoting AGTFP [45]. From a comprehensive perspective, studies have recognized the critical role of agricultural services in the agricultural green transformation. However, few studies have directly focused on the impact of AMS on the agricultural green transformation. At the same time, compared with other agricultural services, the “cross-regional operation” feature of AMS has also been neglected, which will underestimate the spatial spillover effect of AMS in promoting agricultural green transformation. In addition, AMS has multiple supply agents. However, the existing studies have not considered this reality, so there is a lack of in-depth research on the development direction of AMS with the goal of agricultural green transformation.

Given the limitations of existing studies, the research questions of this paper are as follows. (1) Whether there is an impact and spatial spillover effect of AMS on AGTFP. (2) Whether the development direction of different AMS agents is the same under the goal of agricultural green transformation. (3) Is there heterogeneity in the spatial spillover effect of AMS among different regions? To solve these problems, this paper measures the AGTFP using the Super-SBM model based on the panel data of 30 provinces from 2011 to 2020 in China. Based on this, the spatial Durbin model tests the influence of the degree of specialization and large-scale of different AMS supply agents on the AGTFP and the spatial spillover effect. Furthermore, this paper uses the two-regime spatial Durbin model to test the asymmetry of the spatial spillover effect between different regimes. This paper aims to provide theoretical support for the agricultural green transformation in economies with smallholders as the mainstay of agricultural production.

The rest of this paper is organized as follows. Section 2 provides a literature review and constructs a theoretical framework. Section 3 presents the methodology, empirical models, and data sources. Section 4 describes the empirical results and findings. Section 5 provides further discussion and limitations. Section 6 provides the main conclusion.

## 2. Literature Reviews and Theoretical Framework

### 2.1. Literature Reviews

Adam Smith proposed early in the Wealth of Nations that the division of labor could lead to gains from specialization and emphasized that market size determines the division of labor. However, because of the lack of scale, the division of labor in agricultural production tends to be less prevalent than in industrial production. Under this view, Otsuka proposed that only in large farms can investments in mechanization generate adequate returns for operators. Therefore, he argued that the specialized division of labor in Chinese agricultural production would be challenging due to the scale of agriculture and that labor productivity growth would be prolonged [46]. However, the practice of agricultural development in China shows that with an average agricultural size of about 0.5 ha, China has developed a division of labor and experienced rapid agricultural mechanization. For this reason, the rapid development of AMS has facilitated the organic combination of the provision of specialized labor and the services of large harvesting machines, thus effectively increasing agricultural productivity by achieving economies of scale in services [47].

Adequate research has been conducted regarding the factors influencing AMS purchase decisions and the economic and social effects they produce. Specifically, the operation scale is an essential factor influencing the adoption of AMS by farmers. Due to the vast sunk costs and high maintenance costs of agricultural machinery, it is often not cost-effective for smallholders who lack the capital to purchase machinery on their own [17,48]. Therefore, small farmers tend to prefer to purchase AMS. In contrast, large-scale operators tend to purchase machinery on their own and have a tendency to further transform into agricultural mechanization service providers [49]. In addition to the scale of operation, non-farm employment income, education level, and farming experience are influential factors affecting farmers’ purchase of AMS [50,51,52]. Moreover, at the macro level, the level of economic development, population size and agricultural machinery purchase subsidy policy have an important impact on the level of AMS in each province [53]. Some scholars have explored the impact of purchasing AMS on farmers’ agricultural production. Deng et al. found that the productivity of farmers who purchased AMS increased by 25.61% in a sample of farmers in Shandong [54]. A study by Tang et al. showed that agricultural services helped farmers reduce their production costs and technical services had the most significant impact, followed by processing services and AMS [55]. The study by Qiu et al. further indicated that the increase in productivity after purchasing AMS was significantly higher in medium-sized farms than in small and large farms [56]. Some scholars have also explored the impact of AMS adoption on farm household welfare. Among them, Mi et al. used a sample of cotton farmers in Xinjiang and found that the adoption of AMS by small farmers significantly increased household income, consumption expenditure, and off-farm employment opportunities [57]. Lyne et al. found that the extension of AMS in South Africa contributed to higher household agricultural income and helped farmers improve the quality of their agricultural products [58]. However, the breakdown and poor maintenance of agricultural machinery will reduce the profitability of AMS supply agents and further weaken the improvement of AMS to the welfare of farm households [59]. In addition, excessive AMS prices will also lead to the withdrawal of smallholder farmers from agricultural production, which in turn will negatively impact smallholder welfare [60]. In general, existing studies have been valuable and well-explored around AMS. However, a suitable theoretical framework for the mechanism of action between AMS and AGTFP has yet to be established.

### 2.2. Theoretical Framework

By combing the existing studies, this paper constructs a theoretical framework of the impact of AMS on AGTFP from four aspects, including factor allocation, planting structure, technological progress, and spatial spillover effect, as shown in Figure 1. AMS affects AGTFP by changing the allocation of agricultural production factors, mainly resource inputs (labor, land, agricultural machinery) and environmental inputs (pesticides, fertilizers, agricultural films). From the perspective of resource factor inputs, the rapid development of AMS services will directly increase the input scale of agricultural machinery factors, which will directly cause the expansion of agricultural energy consumption and bring potential problems for the green transformation of agriculture in the economy. However, AMS will also effectively promote the transfer of agricultural labor through the labor substitution effect, which expands the land operation scale [48,53]. Furthermore, economies of scale help to achieve increased agricultural productivity and reduce agricultural carbon emissions. In terms of environmental factor inputs, AMS can improve farmers’ expectations of crop yields, which in turn effectively reduces excessive inputs of environmental factors such as pesticides and fertilizers through factor substitution mechanisms [45,61]. Additionally, AMS can help promote the formation of pro-environmental behaviors among farmers, which in turn improves the AGTFP at the micro level [44]. However, it has also been pointed out that AMS enhances the use of abandoned or poorly managed arable land, thereby increasing the intensity of environmental factor inputs such as pesticides [62].

AMS affects AGTFP by changing the cropping structure. Cash crops are more labor-intensive than food crops, and AMS constrains the “non-grain trend “ of crops by facilitating the transfer of agricultural labor. At the same time, food crops are more suitable for mechanization than cash crops, so with the increase in labor cost and the decrease in AMS price, farmers will prefer to grow food crops. In addition, the development of AMS will increase the comparative advantage of food crops by improving the expansion of the agricultural operation scale, thus promoting a “food-based” cropping structure [11]. Compared with cash crops, food crops have a smaller scale of production factor inputs and a more significant carbon sink effect. Therefore, substituting food crops for cash crops will help increase the AGTFP of agricultural systems.

AMS affects AGTFP through technological advances. Firstly, applying AMS can effectively improve the efficiency of all types of energy use in agricultural production, thereby reducing agricultural energy inputs in the input dimension and undesired outputs such as agricultural carbon emissions. Secondly, the rapid development of AMS can also replace traditional agricultural energy with new and renewable energy sources, thus optimizing energy use structure [63]. Thirdly, AMS can bring advanced organizational management experience to agricultural producers and deepen the specialization of agricultural producers through vertical division of labor, which contributes to the improvement of agricultural production organization, management level, and production efficiency [64]. Fourthly, AMS can help increase the timeliness of operations in the agricultural production chain, avoiding the impact of work-hour delays on crop growth, thus indirectly increasing output.

The spatial spillover effect of AMS. Compared with other agricultural services, the cross-regional operation is an important feature of AMS. The vast territory of China and the apparent difference in crop production cycles between regions provide the possibility for the cross-regional operation of AMS. At the same time, the cross-regional operation also helps to expand the market scale and further deepen the degree of vertical division of labor in the whole agricultural system, which is more conducive to the realization of service economies of scale [65]. The cross-regional operations of AMS are mainly driven by large- and medium-sized agricultural machinery. They operate between regions within a day’s economic distance, and operations’ intensity increases yearly [66]. Meanwhile, due to the high seasonal requirements of crop production, AMS agents often operate across regions between different latitudes [67]. Therefore, through cross-regional operations, AMS can influence the allocation of production factors, cropping structure, and technological progress in neighboring regions, changing the AGTFP of the whole agricultural system.

## 3. Methodology and Data

### 3.1. Measurement of AGTFP

AGTFP is an important indicator of the agricultural green transformation. Currently, productivity is widely measured using the non-parametric approach. In a non-parametric method such as envelopment analysis (DEA), the efficiency of a decision-making unit is described by the relationship between inputs and outputs on a linear piecewise frontier constructed by the DEA model [23,68,69,70]. In addition, some scholars have combined DEA models with machine learning to optimize the productivity measures [71,72]. In the early research, the method requires the selection of input and output angles of the model and the same proportional variation of inputs or outputs, thus making it difficult to match its measurement results with the actual situation. Therefore, considering the non-angle and non-radial characteristics of the DEA model, the super-SBM model is proposed. Based on this, this paper incorporates agricultural carbon emissions as non-expected outputs into the super-SBM model to measure the AGTFP. It is assumed that the decision unit k has input vectors $x\in R^{M}$, desired output vectors $y^{g}\in R^{s_{1}}$, and undesired output vectors $y^{b}\in R^{s_{2}}$, respectively. For the decision unit k to be measured, as in Equation (1):(1)$$\rho =\min\frac{1+\frac{1}{m}\sum_{i=1}^{m}\frac{s_{i}^{-}}{x_{\mathrm{ik}}}}{1-\frac{1}{s_{1}{+s}_{2}}\left(\sum_{r=1}^{s_{1}}\frac{s_{r}^{g}}{y_{\mathrm{rk}}^{g}}+\sum_{t=1}^{s_{2}}\frac{s_{t}^{b}}{y_{\mathrm{rk}}^{b}}\right)}\;s.t.\;\overset{n}{\underset{j=1,j\neq k}{\sum }}x_{\mathrm{ij}}\lambda_{j}{-s}_{i}^{-}\leq x_{\mathrm{ik}}\;\overset{n}{\underset{j=1,j\neq k}{\sum }}y_{\mathrm{rj}}\lambda_{j}{+s}_{r}^{g}\leq y_{\mathrm{rk}}^{g}\;\overset{n}{\underset{j=1,j\neq k}{\sum }}y_{\mathrm{tj}}\lambda_{j}{-s}_{t}^{b}\leq y_{\mathrm{tk}}^{b}\;\lambda \geq {0,\;s}^{g}\geq {0,\;s}^{b}\geq {0,\;s}^{-}\geq 0$$
where ρ represents the AGTFP under the super-SBM model, which can be greater than 1, so that the effective decision unit can be distinguished, λ is the weight vector, and $s^{g}$, $s^{b}$, and $s^{-}$ are slack variables. 

From the model set, it is clear that the input and output variables selection is critical for the measurement of AGTFP. Concerning existing studies, this paper constructs an indicator system for measuring AGTFP, as shown in Table 1. In contrast to existing studies, this paper replaces the irrigated area factor with the amount of agricultural water consumption. This is because both irrigated area and land input elements are characterized using land area, and it is unreasonable to put both in the same model. Meanwhile, due to the significant differences in farming practices and crop cultivation structures in different regions, the weight of agricultural water consumption per unit of irrigated area is different. Therefore, it is impossible to truly reflect the amount of water resources input through the irrigated area [22]. In addition, this paper selects agricultural carbon emissions as undesired output. Referring to existing studies [73,74,75,76], this paper defines the sources of agricultural carbon emissions as pesticides, agricultural fertilizers, agricultural plastic films, agricultural diesel fuel, agricultural machinery, and irrigation. The carbon emission factors of each type of carbon source refer to IPCC [77].

### 3.2. Variables Description

#### 3.2.1. Core Explanatory Variables

Based on the two research scales of organization and individual, this study deconstructs the development of AMS into large-scale and specialization, from which four core explanatory variables are constructed. At the organizational level, it mainly includes large-scale AMS supply organizations (LSO) and specialized AMS supply organizations (SPO). The former is indicated by the ratio of the number of AMS organizations with agricultural machinery with the original value of 500,000 yuan or more to the number of total AMS organizations; the ratio of the number of agricultural machinery specialized cooperatives to the number of total AMS organizations indicates the latter. From the individual level, it mainly includes large-scale AMS supply individuals (LSI) and specialized AMS supply individuals (SPI). The former is indicated by the ratio of the number of AMS farmers with agricultural machinery with the original value of 200,000 yuan or more to the number of total AMS farmers; the ratio of AMS specialized farmers to the number of total AMS farmers indicates the latter.

As shown in Figure 2, the development of AMS in China shows the primary trend of large-scale specialization. From the organization level, while the number of AMS organizations steadily increased from 2011 to 2020, LSO continued to grow from 11.51% to 30.40%, and SPO continued to grow from 16.33% to 38.72%. However, at the individual level, while the number of AMS farmers declined, LSI slightly increased from 1.08% to 1.77%, and SPI decreased from 12.45% to 10.53% in 2011–2020. The weakening of AMS at the individual level is due to the massive shift of rural labor, with some AMS farmers moving to non-farm sector employment. It may also be due to the structural shift of some AMS farmers from individual to organizational through cooperatives and acquisitions.

#### 3.2.2. Control Variables

This paper selects the following control variables concerning the existing literature: (1) Degree of industrialization (IND), the ratio of value added of secondary industry to total output value; (2) Per capita disposable income of rural residents (lnDI), in yuan; (3) Agricultural fiscal expenditure (lnAFE), in billion yuan; (4) Area affected, in thousand hectares; (5) Urbanization rate (UR), the ratio of the number of permanent urban residents to the number of permanent residents at the end of the year in the province; (6) GDP per capita (lnGDP), the ratio of gross regional product to the number of permanent residents at the end of the year; (7) Mechanization intensity (MI), the ratio of total agricultural machinery power to sown area; (8) Population (POP), the number of people registered at the end of the year. To exclude the effect of prices, the relevant data are converted to constant prices with 2011 as the base period. Descriptive statistics are shown in Table 2.

### 3.3. Empirical Models

#### 3.3.1. Spatial Durbin Model

In order to study the effects of different AMS supply agents on AGTFP and spatial spillover effects, the following spatial Durbin model (SDM) was constructed for empirical analysis.
(2)$$y_{\mathrm{it}}=\rho \sum_{i=1}^{n}W_{\mathrm{ij}}y_{\mathrm{it}}{+\beta }_{1}{\mathrm{AMS}}_{\mathrm{it}}{+\delta }_{1}\sum_{i=1}^{n}W_{\mathrm{ij}}{\mathrm{AMS}}_{\mathrm{it}}{+\beta }_{2}X_{\mathrm{it}}{+\delta }_{2}\sum_{i=1}^{n}W_{\mathrm{ij}}X_{\mathrm{it}}{+\mu }_{i}{+\gamma }_{t}{+\varepsilon }_{\mathrm{it}}$$

In Equation (2), $y_{\mathrm{it}}$ is the explanatory variable, which indicates the observed value of AGTFP of the province *i* in year *t*; ${\mathrm{AMS}}_{\mathrm{it}}$ is the core explanatory variable, including LSO and SPO at the organizational level and LSI and SPI at the individual level; $X_{\mathrm{it}}$ is the control variable, including the eight control variables mentioned above; $W_{\mathrm{ij}}$ indicates the spatial weight matrix, and in order to ensure the robustness of the research results, the spatial adjacency matrix, geographic distance matrix, and economic distance matrix are constructed for spatial econometric analysis, respectively; ρ is the spatial autocorrelation coefficient, which indicates the impact of AGTFP of neighboring provinces on the local region; $\beta_{1}$ and $\beta_{2}$ are the coefficient to be estimated for the core explanatory variables and control variables, respectively; $\delta_{1}$ and $\delta_{2}$ are the estimation coefficient of the spatial lag term of the core explanatory variables and control variables, respectively; $\mu_{i}$ and $\gamma_{t}$ indicate the individual effect and time effect, respectively, and $\varepsilon_{\mathrm{it}}$ is the random disturbance term.

#### 3.3.2. Two-Regime Spatial Durbin Model

The cross-regional operations of AMS organizations and individuals are driven by demand and revenue and thus exhibit asymmetrical characteristics. On the one hand, the Chinese government set 13 provinces, including Anhui, Shandong, Henan, etc., as the main grain production areas in 2001 to ensure food security. The main grain-production areas undertake important grain production tasks, so their demand for AMS is strong. Meanwhile, developed regions such as Beijing, Shanghai, and Tianjin have higher prices for AMS, which are also more attractive to the flow of AMS organizations and individuals. Therefore, referring to the study of Elhorst [78], this paper constructs a two-regime Spatial Durbin Model to test the heterogeneity effect of AMS to AGTFP. The specific settings of the model are as follows:(3)$$y_{\mathrm{it}}{=\rho }_{d=1}d_{\mathrm{it}}\sum_{i=1}^{n}W_{\mathrm{ij}}y_{\mathrm{it}}{+\rho }_{d=0}{(1-d}_{\mathrm{it}})\sum_{i=1}^{n}W_{\mathrm{ij}}y_{\mathrm{it}}{+\beta }_{1}{\mathrm{AMS}}_{\mathrm{it}}{+\delta }_{1}\sum_{i=1}^{n}W_{\mathrm{ij}}{\mathrm{AMS}}_{\mathrm{it}}{+\beta }_{2}X_{\mathrm{it}}{+\delta }_{2}\overset{n}{\underset{i=1}{\sum }}W_{\mathrm{ij}}X_{\mathrm{it}}{+\mu }_{i}{+\gamma }_{t}{+\varepsilon }_{\mathrm{it}}$$

In Equation (3), $d_{it}$ is a binary dummy variable that distinguishes different regime, and the specific replication rules are as follows. The other coefficients remain consistent with Equation (2).
(4)$$d_{\mathrm{it}}^{1}=\{\begin{array}{l} 1,\;\mathrm{main}\;\mathrm{grain}\;\mathrm{producing}\;\mathrm{area}\;\\ 0,\;\mathrm{else}\; \end{array}$$
(5)$$d_{\mathrm{it}}^{2}=\{\;\begin{array}{l} 1,\;\mathrm{developed}\;\mathrm{area}\;\\ 0,\;\mathrm{else}\; \end{array}$$

### 3.4. Data Resources

Since the SBM model is sensitive to abnormal data and there are differences in the statistical caliber of some regions, the data of Tibet, Hong Kong, Macau, and Taiwan provinces are excluded. Finally, this study constructs the panel data of 30 provinces in China from 2011 to 2020. Specifically, the core explanatory variable AMS is obtained from the *China Agricultural Machinery Industry Yearbook (2012–2021)*; the data on agricultural value added, total agricultural machinery power, crop sown area, agricultural fertilizer application, agricultural plastic film use, agricultural diesel fuel, and pesticide use used in the calculation of AGTFP are taken from the *China Rural Statistical Yearbook (2012–2021)*; the number of employees in the primary industry is obtained from the WIND database; other control variables such as regional GDP and agricultural-related fiscal expenditure are taken from the *China Statistical Yearbook (2012–2021)*.

## 4. Results

### 4.1. Temporal Evolutionary Characteristics of AGTFP in China

Table 3 shows the AGTFP of China’s provinces from 2011 to 2020. Overall, China’s AGTFP shows a stable upward trend, with the average value increasing from 0.1990 in 2011 to 0.5590 in 2020 (line 2), with an average annual growth rate of 13.42%. Between 2011 and 2020, Jilin (line 13) and Inner Mongolia (line 11) have the lowest AGTFP averages of 0.1449 and 0.1549, respectively, and Jilin is also one of the provinces with the slowest AGTFP growth rate of 4.08% per year. Beijing (line 7) and Hainan (line 27) have the highest average value of AGTFP, with 0.6385 and 0.5887, respectively. At the same time, Tianjin (line 8) and Ningxia (line 35) have the fastest growth rate, with an average annual growth rate of 24.28% and 27.22%, respectively.

By 2011–2015 China’s 12th Five-Year Plan period (column 2 and column 6, line 2), the national AGTFP average value increased from 0.1990 to 0.2620 with an average annual growth rate of 7.51%. 2016–2020 is China’s 13th Five-Year Plan period (column 7 and column 11, line 2). The national average value of AGTFP increases from 0.2874 to 0.5590, with an average annual growth rate of 18.15%. In comparison to the “12th Five-Year” period, China’s AGTFP growth rate and magnitude are higher. The reasons for this are, first, the development concept differences. During the “12th Five-Year Plan” period, the main goal of China’s agricultural production was to ensure the adequate supply of food and other major agricultural products. In the “13th Five-Year Plan” period, the basic principle of China’s agricultural production is to adhere to sustainable development, and the “13th Five-Year Plan” clearly proposes to “take the path of output efficiency, resource conservation, and environmental friendliness.” Second, the difference in development basis. During the “12th Five-Year Plan” period, China still faced the difficulties of weak agricultural infrastructure, low material equipment, and lagging social services. At the beginning of the “13th Five-Year Plan”, China built 400 million mu of high-standard farmland, and the contribution rate of agricultural science and technology progress and the total mechanization rate of crop cultivation, planting, and harvesting have reached 56% and 63%, respectively.

From the perspective of different regions, the degree and speed of agricultural green transformation in the main grain production areas are slightly lower than the non-main grain production areas (line 3 and line 4), with the average AGTFP values of 0.1799 and 0.3567 in 2011 to 0.2136 and 0.7138 in 2020, respectively, and the average annual growth rates of 11.61% and 14.81%, respectively. This is because the main grain production areas are critical to ensuring national food security, and increasing total grain production is their top priority. At the same time, this also means that the main grain production areas will be the main battlefield for the agricultural green transformation in China. The mean value of AGTFP in economically developed regions (line 5), represented by large urban agglomerations such as Beijing-Tianjin-Hebei, Yangtze River Delta, and Pearl River Delta, is 0.2510 in 2011 and 0.6812 in 2020, significantly higher than 0.1730 in 2011 and 0.4979 in 2020 in economically developing regions (line 6). Economically developed regions have a higher level of agricultural modernization and a noticeable technology diffusion effect, thus promoting the rapid growth of AGTFP.

### 4.2. Applicability Test of the Spatial Econometric Model

Before using the spatial model regression, the spatial autocorrelation test needs to be performed on AGTFP. As seen from Figure 3, the Moran’s I of AGTFP is between 0.138 and 0.229, and the Moran’s I of each year is significant at the 5% statistical level. It indicates a significant spatial autocorrelation of AGTFP in China, which is suitable for regression analysis using the spatial econometric model. Specifically, during the 12th Five-Year Plan period (2011–2015), the AGTFP of each province in China showed a decreasing trend and fell to the lowest point of 0.138 in 2015. It can be concluded that the inter-provincial AGTFP in China has clustering characteristics, i.e., high values are clustered with high values, and low values are clustered with low values. However, the clustering characteristics vary widely between years and are relatively unstable overall. This also indicates that there are still significant differences in AGTFP between provinces, and there is more room for development.

On this basis, this paper adopts a series of tests to determine the applicability of the spatial Durbin model. First, in the LM test, the LM statistics of the spatial lag model and the spatial error model are 146.754 and 229.790, respectively, and both are significant at the 1% statistical level. This indicates that the original hypothesis of no spatial lag and the spatial error should be rejected, and the spatial panel model should be used for the empirical analysis. Second, in the LR test, the statistical value is 47.860 when the spatial Durbin model is compared with the spatial error model and 31.360 when the spatial Durbin model is compared with the spatial lag model, and both are significant at the 1% statistical level. This indicates that the spatial Durbin model cannot be reduced to a spatial error model and a spatial lag model in this study. Finally, the Hausman test has a statistical value of 29.120, which is significant at the 1% statistical level. This indicates that the original hypothesis of using random effects should be rejected, and a fixed effects spatial Durbin model should be used.

### 4.3. The Impact of AMS Supply Organizations on AGTFP

Table 4 demonstrates the effect of AMS supply organization on AGTFP. Under the three spatial weight matrices, the spatial autoregressive coefficient of AGTFP is positive and significant at the 1% statistical level for both LSO and SPO as the core explanatory variables, which is consistent with the studies of Xiao et al. [23] and Ma et al. [79]. This suggests that the AGTFP of neighboring provinces or provinces with similar degrees of economic development can effectively contribute to the increase in AGTFP of the local province. The possible influence mechanism is the inter-regional technology spillover effect and demonstration effect. At the same time, this finding is consistent with the previous spatial autocorrelation test and further reflects the validity of the findings of this study.

Regarding the main effects, the regression coefficients of LSOs ranged from 0.132–0.172, all of which were statistically significant at the 10% level. The regression coefficients of SPO ranged from 0.194–0.255, all of which were significant at the 1% statistical level. These results show that increasing the scale and specialization of AMS organizations in the local region can effectively increase AGTFP and thus promote the green transformation of agriculture. Specifically, the expansion of the AMS organization scale helps to increase agricultural productivity and effectively restrain excessive input of environmental factors through the factor substitution mechanism, thus increasing AGTFP. At the same time, the increase in AMS organization specialization helps to accelerate the technological progress in agricultural production, promoting agricultural green transformation.

Regarding the control variables, the expansion of the affected area will lead to a decrease in food production and agricultural output, resulting in a significant decrease in AGTFP. As for the micro-level explanation, after suffering from natural disasters, in order to reduce losses to avoid falling into poverty, farmers and other operators will increase factor inputs such as fertilizers and pesticides on non-affected plots, which increases undesired output. Meanwhile, higher regional GDP means better regional economic development and agricultural infrastructure, promoting a significant increase in AGTFP. This finding also provides supporting evidence for the later paper to explore the effect of AMS on AGTFP from the level of economic development. In addition to this, the intensity of agricultural mechanization and population size will also significantly increase AGTFP, which is consistent with Zhu et al. [31] and Chi et al. [80].

In terms of the spatial lagged variables, the regression coefficients of LSO*W ranged from −0.437 to −1.662 and were all significant at the 5% statistical level, i.e., there was a significant negative spatial spillover effect of the effect of LSO on AGTFP. On the other hand, the regression coefficients of SPO*W ranged from 0.379 to 1.265, and all were significant at the 1% statistical level, i.e., there was a significant positive spatial spillover effect of the effect of SPO on AGTFP. The spatial spillover effect is significant precisely because both LSOs and SPOs are the supplying agents of agricultural machinery cross-regional operation services. In addition, the regression coefficient of lnAFE*W is significantly positive, indicating that the expansion of the scale of agriculture-related expenditures in neighboring provinces will contribute to the increase in AGTFP in this province, a similar finding to that of Xiao et al. [23].

On this basis, the spatial effects of the core explanatory variables are further decomposed, and the results are shown in Table 5. Taking the spatial adjacency weight matrix (W1) as an example, the direct effects of LSO and SPO on AGTFP are significantly positive at the 5% and 1% statistical levels, respectively, which indicate that the scale development of AMS supply organizations can effectively promote the increase in AGTFP in this province. However, the indirect effect of LSO is significantly negative at the 1% statistical level, indicating that the scaling up of AMS supply organizations in neighboring provinces will inhibit the increase in AGTFP in this province. In contrast, the indirect effect of SPO is statistically significant and positive at the 5% level, indicating that the increased specialization of AMS supply organizations in neighboring provinces will promote the increase in AGTFP in this province. The indirect effects are closely related to the differences in the production and operation modes between LSOs and SPOs. The cross-regional operation of AMS supply organizations is a supplement to the supply of AMS in neighboring provinces based on meeting the demand for AMS in the local province. It forms a complementary relationship with the AMS supply bodies in neighboring provinces. However, with the further expansion of the scale of AMS supply organizations, LSOs mainly involving agricultural enterprises and leading enterprises are formed, equipped with higher-value agricultural machinery and more advanced production technologies. Cross-regional operations of LSOs can achieve a monopoly of the agricultural machinery market in neighboring provinces with cost and price advantages [81], thus causing a severe impact on the agricultural mechanization service system in foreign provinces and ultimately reducing the AGTFP. At the same time, the cross-regional operation of LSOs will also form an AMS system with large-scale households as the core through the coercive effect of service economies of scale. It will accelerate the marketization process of land leasing and force smallholders out of agricultural production [49,82].

### 4.4. The Impact of AMS Supply Individuals on AGTFP

As shown in Table 6, the regression coefficients of LSI ranged from 0.224 to 0.386 under the three spatial weight matrices, and all were statistically significant at the 10% level. This indicates that the scaling of AMS individuals helps promote agricultural green transformation. Meanwhile, the regression coefficient of LSI is higher than that of LSO. This is because the unique advantage of LSI over LSO is that it is nested in the social network of rural society as a member of Chinese rural society. Therefore, LSI can effectively improve the pro-environmental behavior of farmers through the moderating effect of social trust, which in turn can increase AGTFP more efficiently [83]. The regression coefficients of SPI range from −0.398 to −0.311, all were significant at the 1% statistical level. This suggests that the degree of specialization of AMS individuals will inhibit the green transformation of agriculture. There are three main reasons for this. First, the formation of economies of scale in service is the essential condition for AMS to promote the green transformation of agriculture in the economy. However, increasing AMS individual specialization does not mean a simultaneous scale increase. As shown by statistical data, the number of SPI in China in 2020 is 5.96 times that of LSI, and the value is as high as 330.036 in Guizhou province. Second, specialized farmers take the provision of AMS as their main source of income and thus may cause excessive input of agricultural machinery elements driven by profit maximization, increasing the undesired output of agricultural production. Third, the value of agricultural machinery owned by specialized farm households is lower than that of LSI. Due to their endowment constraints, they cannot afford to carry out frequent renewal and maintenance of agricultural machinery. Thus, their supply quality of AMS is lower, which in turn causes a decrease in the efficiency of agricultural production factor allocation. In terms of spatial lag variables, the regression coefficients of LSI*W and SPI*W are not significant, which indicates that there is no spatial spillover effect of the effects of LSI and SPI on AGTFP.

This paper further decomposes the spatial effects of LSI and SPI, and the results are shown in Table 7. Under the three spatial weight matrices, the direct effect of LSI is significantly positive at the 10% statistical level, and the direct effect of SPI is significantly negative at the 1% statistical level. This indicates that the scale development of individual AMS significantly increases the AGTFP of the local province, while specialization development suppresses the AGTFP of the local province. Meanwhile, the indirect effects of LSI and SPI are insignificant, which means the LSI and SPI of neighboring provinces do not affect the AGTFP of the local province. The reason for this is that, compared to AMS supply organizations, AMS supply individuals face the dilemma of higher acquisition and maintenance costs of agricultural machinery, insufficient demand matching ability due to information asymmetry, and difficulties in transporting agricultural machinery across regions [65]. Therefore, it is difficult for AMS supply individuals to influence the green transformation of agriculture in other provinces through cross-regional operations.

### 4.5. Regional Heterogeneity Analysis

Table 8 demonstrates the empirical results of the two-regime spatial Durbin model based on the spatial weight matrix of economic distance. The comparison with the results in Table 4 and Table 6 reveals that the significance and the direction of action of the regression coefficients of each core explanatory variable remain consistent, which further strengthens the reliability and stability of the findings of this study. On this basis, we focus on the asymmetry of spatial spillover effects among different zone systems. From the results of $d^{1}$, the regression coefficient of rho1 is negative but not significant, the regression coefficient of rho2 is significantly positive at the 1% statistical level, and the regression coefficient of rho1-rho2 is significantly negative at the 5% statistical level. This indicates an asymmetry in the spatial spillover effect of AGTFP, and the spatial spillover effect is significantly higher in non-main grain production areas than in main grain production areas. There are two main reasons for this. First, the crop sowing area in the main grain production areas is vast, and the terrain is flat, so the demand for AMS is more significant and more favorable for agricultural machinery operation. With the rapid development of AMS, the main grain production areas with massive market scale can easily form service scale economy and operation scale economy through the horizontal and vertical division of labor, significantly improving AMS supply agents’ profit. Therefore, the AMS supply agents in the main grain production areas are more inclined to prioritize meeting the demand for AMS in the local province. Second, from 2011 to 2020, the average value of SPO in non-main grain production areas increased from 0.2430 to 0.5370, and the average value of LPO increased from 0.1514 to 0.3941, while the average value of SPO in grain-producing regions increased from 0.2634 to 0.4536, and the average value of LPO increased from 0.1967 to 0.3542. Therefore, compared with non-main grain production areas, the average value of LPO increased from 0.1514 to 0.3941. Although they have more total AMS supply organizations, main grain production areas do not show significant advantages in developing AMS specialization and large-scale and even gradually lag from the early lead.

From the results of $d^{2}$, the regression coefficients of rho1 are all significantly positive at the 1% statistical level, the regression coefficients of rho2 are not significant, and the regression coefficients of rho1-rho2 are all significantly positive at the 5% statistical level. This indicates that the spatial spillover effect of AGTFP is asymmetric among provinces with different levels of economic development, and the spatial spillover effect is significantly higher in economically developed provinces than in other provinces. This is because economically developed provinces have the advantage of transportation infrastructure, which helps to promote the free flow of factors between regions [84,85]. Therefore, it helps to reduce the mechanical transportation costs of AMS supply organizations when operating across regions, thus forming an asymmetry of the spatial spillover effect. In addition to this, compared with other provinces, economically developed provinces have a higher level of scientific research investment and promotion of agricultural machinery [86], which provides essential support for AMS to exert spatial spillover effects.

## 5. Discussion

For countries constrained by the pattern of “smallholder management” due to the fragmentation of land and unclear land property rights, attempts to rely on economies of scale to deepen the horizontal division of labor in agriculture and thus promote the agricultural green transformation are often ineffective. In contrast, the construction of AMS system can deepen the vertical division of labor in agriculture through the realization of the service scale economy and thus become another effective path to agricultural green transformation in economies with smallholders as the mainstay of agricultural production. This paper’s theoretical contributions and innovations are mainly reflected in the following three aspects by comparing the existing studies.

### 5.1. Development Path of AMS under the Goal of Agricultural Green Transformation

The existing studies have demonstrated the critical role of AMS in improving agricultural productivity and environmental efficiency at both macro and micro levels [42,44,54,55,57,83]. These findings provide an essential theoretical basis for AMS’s positive role in agriculture’s green transformation. However, the issue of how to promote the rapid development of AMS remains controversial. The reason for this is that most studies have measured the development of AMS in terms of AMS production value and farmers’ decision to purchase AMS. The resulting policy insights are focused on two general aspects of promoting the rapid development of AMS and improving farmers’ responsiveness, which is challenging to form practical guidance for the rapid development of AMS to achieve the goal of agricultural green transformation. Unlike the existing studies, this paper constructs indicators related to the development of AMS in terms of specialization and the degree of scale from the multiple service supply agents in the Chinese AMS market. The results show that the impact of different AMS supply agents on AGTFP shows apparent differentiation. Among them, the impact of SPO, LSO, and LSI on AGTFP are significantly positive, but the effects of SPI on AGTFP are significantly negative. This key finding provides essential insights into the development path of China’s AMS system under the goal of green transformation in agriculture, namely, to increase the support of AMS supply organizations and actively promote the transformation of AMS supply individuals.

Expressly, compared to AMS supply organizations, AMS supply individuals are limited by their inadequate endowments, which make it difficult for them to acquire high-value agricultural machinery and have relatively low management skills due to the lack of training [87]. Meanwhile, although China has been implementing the agricultural machinery acquisition subsidy policy for a long time, the beneficiaries of this policy are mainly large-scale operators, not small-scale operators [88]. Therefore, due to endowment constraints and policy exclusion, it is difficult for AMS supply individuals to provide high-quality AMS with green agricultural attributes. Based on this, actively promoting the transformation of AMS supply individuals becomes the primary issue in constructing the development path of AMS. This paper presents two transformation possibilities combined with China’s agricultural development practice. First, with the rapid rise of labor prices, the wage income that can be brought by non-farm employment exceeds the operating income by providing AMS. The rapid labor market development provides an opportunity for the transformation of AMS supply individuals. Therefore, it is possible to promote the withdrawal of some agricultural machinery professionals from the AMS market by increasing the non-agricultural transfer of rural labor. This initiative can eliminate some old and low-value agricultural machinery to improve the quality of AMS supply and give up part of the AMS market demand to cultivate other AMS supply agents. Secondly, farmers’ cooperatives have played an essential role in China’s rural economic development by building a close interest linkage mechanism among smallholders through the “weak-weak association” [89,90]. Similarly, agricultural machinery professional cooperatives are an essential part of the AMS supply organization and have played an important role in promoting the agricultural green transformation. Therefore, AMS supply individuals with high-value agricultural machinery can be actively fostered to transform into AMS supply organizations in a “weak-weak association” approach.

### 5.2. Characteristics of AMS Cross-Regional Operations

There has been a debate on the relationship between agricultural mechanization and the green transformation of agriculture. Specifically, some scholars believe that the negative environmental impact of energy consumption caused by agricultural mechanization cannot be ignored. In contrast, some scholars believe that the carbon reduction effect of agricultural mechanization will offset or even reverse its negative environmental impact. The main reason for this debate is that some existing studies ignore the critical impact mechanism of cross-regional operation of agricultural mechanization [42,80]. Meanwhile, the impact of AMS on AGTFP will be underestimated because AMS operating across regions is not counted in the total AMS of neighboring provinces. With the promotion of agricultural mechanization, the scale of cross-regional operation of AMS in China has become larger and larger. In 2020, the area of cross-regional operation of agricultural machinery in China was 19899.67 khm^2^, accounting for 35.22% of the area operated by agricultural machinery cooperatives. Therefore, the cross-regional operation has become an essential channel for AMS to promote agricultural green transformation. Based on this, this paper uses a spatial Durbin model test to find a significant spatial spillover effect of AMS on AGTFP. This finding further demonstrates the critical reality that AMS promotes agricultural green transformation in neighboring provinces through cross-regional operations.

The analysis of the spatial lag term found that the spatial spillover effect of AMS supply organizations was significant. However, the spatial spillover effect of AMS supply individuals was not significant. This finding suggests that the cross-regional operation of AMS is mainly driven by AMS supply organizations, similar to the findings of Fang et al. [66] and Wu et al. [67]. Unlike existing studies, this paper further finds that the spatial spillover effect of specialization of AMS supply organizations on AGTFP is significantly positive. In contrast, the spillover effect of large-scale AMS supply organizations is significantly negative. This finding suggests that a single-minded push for rapid AMS development does not effectively promote a green transformation of agriculture. Local government should regulate the cross-regional operation behavior of large-scale AMS supply organizations is necessary. Furthermore, it is also necessary to cultivate AMS supply agents in the local province through agricultural machinery purchase subsidy policies. These initiatives can counteract the impact of large-scale AMS supply organizations from other regions on the agricultural mechanization service market in the local province.

## 6. Conclusions

This study examines the development of AMS in China and explores how economies with smallholder as the mainstay of agricultural production can construct a transition path to green agriculture. Based on the panel data of 30 provinces in China from 2011 to 2020, this paper measures the AGTFP using the Super-SBM model. Based on this, using the spatial Durbin model and the two-regime spatial Durbin model, this paper tests the influence and spatial effect of different AMS supply agents on AGTFP. The main conclusions are as follows.

First, during the study period, China’s AGTFP showed a stable growth trend, with the mean value increasing from 0.1990 in 2011 to 0.5590 in 2020. Among them, the AGTFP of Beijing and Hainan is relatively higher. In contrast, the AGTFP of Jilin and Inner Mongolia are always lower. By stage, compared with the 12th Five-Year Plan period (2011–2015), the average value and the growth rate of AGTFP in the 13th Five-Year Plan period (2016–2020) is higher and faster. By region, agricultural green transformation is faster in non-main grain production and economically developed regions. In contrast, agricultural green transformation tasks are relatively heavier in food-producing and less economically developed regions.

Second, AMS is an essential factor influencing the growth of AGTFP, but there are still differences among different supplying agents. At the organizational level, the degree of specialization and large-scale of AMS supply organizations will significantly contribute to the increase in AGTFP of the local province. However, the cross-regional operation of specialized AMS supply organizations will significantly increase the AGTFP of neighboring provinces. In contrast, the cross-regional operation of large-scale AMS supply organizations will significantly decrease the AGTFP of neighboring provinces. At the individual level, the degree of specialization of AMS supply individual will significantly decrease the AGTFP of the local province. In contrast, the large-scale MAS supply individual has the opposite effect. At the same time, individual AMS supply agents cannot operate across regions.

Third, the spatial spillover effect of AGTFP is asymmetric due to the significant differences in the supply of AMS among different regions. Specifically, the spatial spillover effect of AGTFP is significantly higher in non-main grain production areas than in main grain production areas. In comparison, the spatial spillover effect of AGTFP is significantly higher in economically developed areas than in less developed areas. Meanwhile, the spatial spillover effect asymmetry also reflects the flow direction of AMS resources.

Based on the above conclusions, to further promote the agricultural green transformation in China, this paper provides the following policy implications. First, promoting the popularization of agricultural mechanization throughout agricultural production. It will help to provide an essential foundation for the rapid development of the AMS market. Meanwhile, with the goal of green transformation of agriculture, it is necessary to enhance agricultural machinery’s R&D and production capacity and promote clean energy-based agricultural machinery to replace traditional fossil energy-based agricultural machinery. Second, actively promote the transformation of AMS supply individuals to AMS supply organizations. Encourage small farmers to join together extensively to realize the transformation through the “farmers’ cooperative.” It can improve the organization of AMS supply subjects. Third, improve the service quality of AMS supply organizations. Increase the agricultural machinery purchase subsidies for AMS supply organizations to improve the service quality and agricultural machinery scale of AMS supply organizations. Meanwhile, strengthen the construction of agricultural mechanization personnel to improve the management capacity of AMS supply organizations. Fourth, the main grain-producing and less economically developed regions should accelerate the construction of local AMS systems. The central government needs to provide corresponding financial support, resource inclination, and legal regulation to guarantee the construction of the AMS systems in critical regions. It will help to avoid the unbalanced and insufficient process of agricultural green transformation among regions.

The main limitations of this paper are the following three aspects. First, based on provincial panel data, this study finds a spatial spillover effect of the cross-regional operation behavior of AMS supply organizations on the impact of AGTFP. However, although AMS supply individuals cannot operate across regions between provinces, it is still being determined whether they operate across regions between smaller administrative units such as counties and communes. Therefore, future research can explore more deeply the cross-regional operation characteristics of different AMS supply individuals based on county panel data. Second, combining statistical data and existing studies [56,65], AMS in China has developed rapidly since 2004. Unfortunately, limited by data availability issues, this study only examined the relationship between AMS and AGTFP during 2010–2020 and has yet to provide a clear picture of the full development of AMS in China. Therefore, longer-term panel data could be considered in future studies to more accurately capture the relationship between the development of AMS and the agricultural green transformation. Third, AMS exists in many aspects of agricultural production, such as tillage, sowing, irrigation, and harvesting [43,64]. A possible future research direction is to explore in depth the heterogeneous role of AMS in different product segments in promoting the agricultural green transformation, which will help to construct a more comprehensive AMS system.

## Figures and Tables

**Figure 1 ijerph-20-01655-f001:**
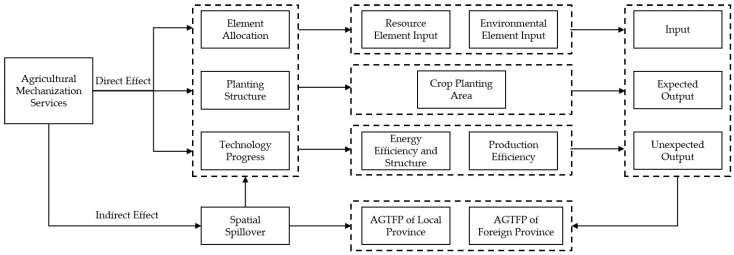
The theoretical Framework for the impact of AMS on AGTFP.

**Figure 2 ijerph-20-01655-f002:**
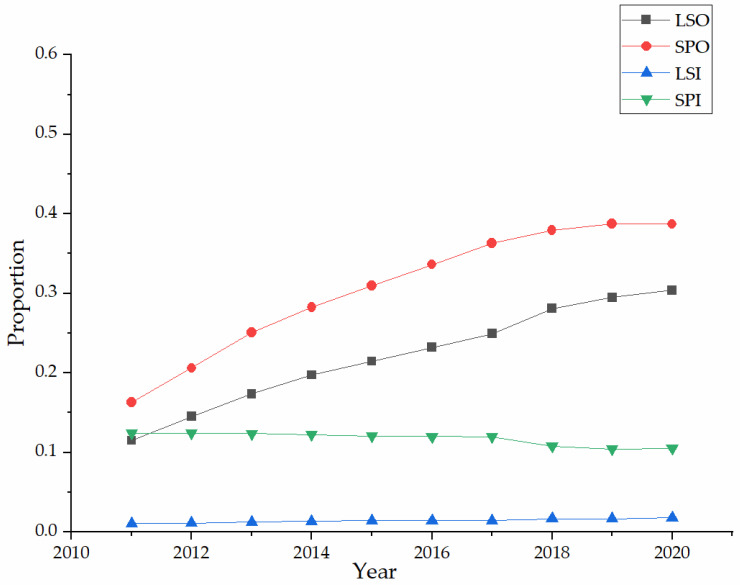
Development of agricultural mechanization services.

**Figure 3 ijerph-20-01655-f003:**
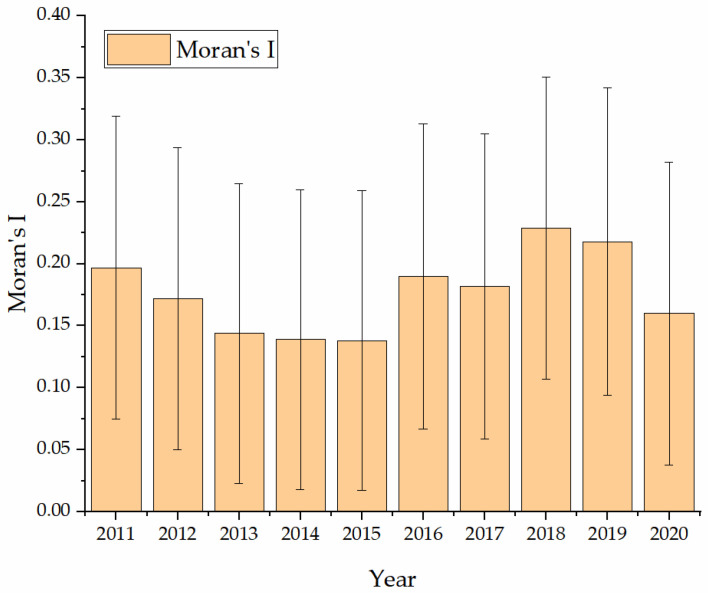
Moran’s I of Agricultural Green Total Factors Productivity in China, 2011 to 2020.

**Table 1 ijerph-20-01655-t001:** Indicator system construction for AGTFP.

Indicator	Index	Definition	Mean	Unit
Input indicators	Labor input	Number of employees in agriculture	780.16	10,000 peoples
	Land input	The sowing area of crops	5336.93	1000 HA
	Mechanical input	Total power of agricultural machinery	3318.60	10,000 kW
	Fertilizer input	Chemical fertilizer application in agriculture	185.89	10,000 tons
	Film input	Amount of Agricultural plastic film application	7.98	10,000 tons
	Pesticide input	Amount of pesticide application	5.35	10,000 tons
	Energy input	Amount of Agricultural diesel consumption	66.75	10,000 tons
	Water input	Amount of Agricultural water consumption	121.89	100 million m^3^
Output indicators	Desired output	The gross production of agriculture	1784.42	100 million yuan
	Undesired output	Agricultural carbon emissions	29,416.66	10,000 tons

**Table 2 ijerph-20-01655-t002:** Descriptive statistics of control variables.

Variable	Mean	Std. Dev.	Max	Min
IND	0.337	0.080	0.100	0.574
lnDI	9.354	0.418	8.271	10.461
lnAFE	6.160	0.573	4.519	7.200
lnDIS	5.985	1.550	0.693	8.349
UR	59.006	12.218	35.030	89.600
lnGDP	10.841	0.436	9.706	12.013
MI	0.328	0.112	0.136	0.693
POP	4599.783	2837.845	568.000	12,624.000

**Table 3 ijerph-20-01655-t003:** The measurement results of AGTFP in China, 2011–2020.

Province	2011	2012	2013	2014	2015	2016	2017	2018	2019	2020
Whole	0.1990	0.2168	0.2353	0.2489	0.2620	0.2874	0.3094	0.3534	0.4327	0.5590
MGP	0.1799	0.1938	0.2083	0.2157	0.2229	0.2337	0.2416	0.2572	0.2827	0.3567
NMGP	0.2136	0.2344	0.2560	0.2743	0.2919	0.3285	0.3612	0.4269	0.5474	0.7138
EDR	0.2510	0.2733	0.2989	0.3154	0.3346	0.3684	0.4026	0.4595	0.5887	0.6812
UEDR	0.1730	0.1885	0.2035	0.2157	0.2257	0.2469	0.2628	0.3003	0.3547	0.4979
Beijing	0.3095	0.3768	0.4432	0.4943	0.5720	0.6139	0.6896	0.8063	1.0572	1.0217
Tianjin	0.1607	0.1694	0.1829	0.1935	0.2013	0.2087	0.2319	0.3130	0.3422	0.4642
Hebei	0.1655	0.1779	0.1935	0.1894	0.1859	0.2014	0.2142	0.2430	0.2602	0.3007
Shanxi	0.1366	0.1451	0.1559	0.1628	0.1564	0.1738	0.1937	0.2029	0.2180	0.2582
Inner Mongolia	0.1388	0.1457	0.1567	0.1558	0.1496	0.1544	0.1460	0.1561	0.1660	0.1803
Liaoning	0.2074	0.2290	0.2400	0.2431	0.2689	0.2772	0.2851	0.3235	0.3659	0.3925
Jilin	0.1445	0.1521	0.1529	0.1525	0.1470	0.1305	0.1199	0.1332	0.1429	0.1737
Heilongjiang	0.1418	0.1726	0.2123	0.2251	0.2234	0.2261	0.2399	0.2536	0.2823	0.3050
Shanghai	0.3783	0.3894	0.4177	0.4117	0.4089	0.3600	0.3548	0.4444	0.4600	0.4356
Jiangsu	0.2197	0.2442	0.2566	0.2678	0.2994	0.3032	0.3100	0.3119	0.3283	0.3973
Zhejiang	0.2261	0.2447	0.2791	0.3005	0.3139	0.3598	0.4334	0.4496	0.5377	0.5467
Anhui	0.1330	0.1431	0.1506	0.1615	0.1623	0.1690	0.1820	0.1840	0.1978	0.2169
Fujian	0.2892	0.3233	0.3445	0.3760	0.3976	0.5018	0.5953	0.7180	1.0882	1.0268
Jiangxi	0.1290	0.1380	0.1822	0.1910	0.2117	0.2303	0.2339	0.2466	0.2695	0.2887
Shandong	0.1936	0.1999	0.2262	0.2402	0.2495	0.2531	0.2591	0.2767	0.2929	0.3163
Henan	0.1897	0.2002	0.2066	0.2297	0.2290	0.2353	0.2434	0.2652	0.2944	0.3475
Hubei	0.2231	0.2379	0.2488	0.2527	0.2573	0.2907	0.3042	0.3149	0.3530	0.3912
Hunan	0.2003	0.2080	0.2030	0.2059	0.2061	0.2203	0.2322	0.2401	0.2880	0.3265
Guangdong	0.2918	0.3120	0.3413	0.3557	0.3733	0.4705	0.5106	0.5865	0.7635	1.1907
Guangxi	0.2183	0.2233	0.2348	0.2442	0.2541	0.2795	0.3054	0.3301	0.4389	0.4881
Hainan	0.3048	0.3362	0.3490	0.3946	0.4270	0.5433	0.5923	0.8344	1.0507	1.0547
Chongqing	0.2178	0.2355	0.2488	0.2616	0.2725	0.3220	0.3372	0.3732	0.6644	1.0216
Sichuan	0.2528	0.2702	0.2785	0.2898	0.3076	0.3461	0.3712	0.3951	0.4344	1.0000
Guizhou	0.1585	0.1984	0.2248	0.2934	0.3967	0.4520	0.5047	0.5756	0.7185	1.0977
Yunnan	0.1424	0.1600	0.1794	0.1899	0.1915	0.2016	0.2115	0.2824	0.4057	1.0000
Shaanxi	0.2495	0.2671	0.2943	0.3203	0.3210	0.3548	0.3812	0.4548	0.5889	1.0372
Gansu	0.1008	0.1087	0.1158	0.1184	0.1229	0.1403	0.1629	0.1803	0.2065	0.3193
Qinghai	0.1539	0.1743	0.2014	0.2057	0.1989	0.2219	0.2342	0.2579	0.3168	0.4640
Ningxia	0.1197	0.1285	0.1402	0.1468	0.1633	0.1787	0.1902	0.2167	0.2102	0.3456
Xinjiang	0.1727	0.1924	0.1984	0.1935	0.1912	0.2021	0.2116	0.2308	0.2387	0.3620

Note: MGP is the main grain production area; NMGP is the non-main grain production area; EDR is the economically developed region; NEDR is the economically developing region.

**Table 4 ijerph-20-01655-t004:** The impact of AMS on AGTFP in China: the organizational level.

	$W_{1}$		$W_{2}$		$W_{3}$	
LSO	0.164 * (0.090)		0.132 * (0.071)		0.172 ** (0.083)	
SPO		0.228 *** (0.057)		0.255 *** (0.077)		0.194 *** (0.064)
IND	−0.703 ** (0.322)	−0.099 (0.303)	−0.432 (0.311)	−0.018 (0.292)	−0.508 * (0.308)	−0.006 (0.286)
lnDI	−0.070 (0.208)	−0.309 (0.204)	−0.192 (0.200)	−0.351 * (0.194)	−0.240 (0.203)	−0.429 ** (0.195)
lnAFE	−0.076 (0.049)	−0.077 (0.050)	−0.054 (0.048)	−0.032 (0.047)	−0.093 * (0.050)	−0.055 (0.050)
lndisaster	−0.015 * (0.009)	−0.016 * (0.009)	−0.015 * (0.008)	−0.016 ** (0.008)	−0.018 ** (0.008)	−0.232 *** (0.008)
UR	0.000 (0.005)	0.006 (0.005)	−0.001 (0.005)	−0.001 (0.005)	0.000 (0.005)	0.002 (0.005)
lnGDP	0.303 *** (0.075)	0.261 *** (0.073)	0.288 *** (0.071)	0.247 *** (0.064)	0.255 *** (0.069)	0.230 *** (0.064)
MI	0.321 *** (0.117)	0.297 *** (0.112)	0.320 *** (0.110)	0.300 *** (0.106)	0.377 *** (0.113)	0.337 *** (0.110)
POP	0.000 ** (0.000)	0.000 * (0.000)	0.000 *** (0.000)	0.000 ** (0.000)	0.000 *** (0.000)	0.000 *** (0.000)
LSO*W	−0.437 *** (0.167)		−1.662 ** (0.673)		−1.119 ** (0.507)	
SPO*W		0.379 *** (0.146)		1.265 *** (0.639)		0.877 ** (0.402)
IND*W	0.433 (0.643)	0.560 (0.637)	−3.737 (2.823)	−1.859 (2.730)	2.164 (2.141)	0.412 (2.106)
lnDI*W	0.086 (0.382)	0.279 (0.369)	0.701 (1.229)	1.044 (1.135)	0.095 (0.887)	0.605 (0.791)
lnAFE*W	0.315 *** (0.098)	0.311 *** (0.099)	1.479 *** (0.357)	1.808 *** (0.344)	0.287 (0.231)	0.470 ** (0.201)
lndisaster*W	−0.006 (0.017)	0.010 (0.016)	0.028 (0.054)	0.734 (0.055)	−0.022 (0.034)	0.030 (0.032)
UR*W	−0.045 *** (0.011)	−0.047 *** (0.011)	−0.027 (0.029)	−0.022 (0.028)	−0.006 * (0.021)	−0.016 (0.019)
lnGDP*W	0.224 * (0.134)	0.357 *** (0.110)	−0.196 (0.529)	0.571 (0.407)	−0.062 (0.389)	0.510 (0.333)
MI*W	−0.091 (0.232)	0.135 (0.225)	0.471 (0.695)	0.767 (0.680)	−0.852 (0.689)	0.468 (0.630)
POP*W	0.000 * (0.000)	0.000 (0.000)	0.000 (0.000)	0.000 (0.000)	0.000 (0.000)	0.000 (0.000)
rho	0.232 ** (0.093)	0.289 *** (0.090)	1.638 *** (0.294)	1.845 *** (0.283)	0.728 *** (0.209)	0.802 *** (0.204)
Province FE	YES	YES	YES	YES	YES	YES
Time FE	YES	YES	YES	YES	YES	YES
R-squared	0.261	0.235	0.444	0.511	0.245	0.373
Obs	300	300	300	300	300	300

Note: Standard errors in parentheses, *, ** and *** indicate significant at the 10%, 5% and 1% statistical levels, respectively.

**Table 5 ijerph-20-01655-t005:** Spatial effect decomposition: the organizational level.

	$W_{1}$			$W_{2}$			$W_{3}$		
	DE	IE	TE	DE	IE	TE	DE	IE	TE
LSO	0.192 ** (0.094)	−0.412 *** (0.153)	−0.219 (0.150)	0.232 ** (0.096)	−0.825 *** (0.313)	−0.593 ** (0.282)	0.270 *** (0.070)	−1.270 *** (0.439)	−1.000 ** (0.445)
SPO	0.211 *** (0.068)	0.278 ** (0.123)	0.489 *** (0.157)	0.209 *** (0.067)	0.355 (0.233)	0.564 ** (0.259)	0.172 *** (0.065)	0.442 * (0.239)	0.614 (0.254)

Note: Standard errors in parentheses, *, ** and *** indicate significant at the 10%, 5% and 1% statistical levels, respectively.

**Table 6 ijerph-20-01655-t006:** The impact of AMS on AGTFP in China: the individual level.

	$W_{1}$		$W_{2}$		$W_{3}$	
LSI	0.224 * (0.130)		0.386 ** (0.164)		0.237 * (0.138)	
LSI*W	−0.273 (0.252)		−0.523 (0.687)		0.110 (0.156)	
SPI		−0.398 *** (0.102)		−0.311 *** (0.110)		−0.341 *** (0.101)
SPI*W		0.250 (0.274)		1.290 (0.854)		−0.093 (0.405)
rho	0.295 *** (0.091)	0.258 *** (0.092)	1.813 *** (0.284)	1.826 *** (0.291)	0.841 *** (0.205)	0.786 *** (0.215)
Control Variables	YES	YES	YES	YES	YES	YES
Province FE	YES	YES	YES	YES	YES	YES
Time FE	YES	YES	YES	YES	YES	YES
R-squared	0.185	0.293	0.512	0.508	0.451	0.467
Obs	300	300	300	300	300	300

Note: Standard errors in parentheses, *, ** and *** indicate significant at the 10%, 5% and 1% statistical levels, respectively.

**Table 7 ijerph-20-01655-t007:** Spatial effect decomposition: the individual level.

	$W_{1}$			$W_{2}$			$W_{3}$		
	DE	IE	TE	DE	IE	TE	DE	IE	TE
LSI	0.238 * (0.134)	−0.270 (0.233)	−0.032 (0.250)	0.414 ** (0.171)	−0.485 (0.416)	−0.071 (0.429)	0.241 * (0.143)	0.035 (0.117)	0.277 * (0.152)
SPO	−0.406 *** (0.105)	0.290 (0.251)	−0.116 (0.265)	−0.360 *** (0.109)	0.906 (0.565)	0.546 (0.485)	−0.336 *** (0.104)	0.041 (0.305)	−0.296 (0.330)

Note: Standard errors in parentheses, *, ** and *** indicate significant at the 10%, 5% and 1% statistical levels, respectively.

**Table 8 ijerph-20-01655-t008:** Empirical results of heterogeneity analysis.

	$d^{1}$				$d^{2}$			
LSO	0.171 *** (2.612)				0.173 *** (2.587)			
LSO*W	−0.180 ** (−2.140)				−0.183 ** (−2.201)			
SPO		0.135 * (1.886)				0.131 * (1.746)		
SPO*W		0.308 * (1.866)				0.286 * (1.730)		
LSI			0.546 *** (3.281)				0.597 *** (3.526)	
LSI*W			−0.035 (−0.033)				0.318 (0.305)	
SPI				−0.234 *** (−2.728)				−0.223 *** (−2.584)
SPI*W				0.441 (1.159)				0.439 (1.132)
rho1	−0.157 (−1.250)	−0.177 (−1.405)	−0.103 (−0.813)	−0.167 (−1.298)	0.272 *** (2.982)	0.269 *** (2.972)	0.272 *** (2.947)	0.272 *** (2.947)
rho2	0.410 *** (3.303)	0.405 *** (3.268)	0.348 *** (2.776)	0.411 *** (3.289)	−0.197 (−0.986)	−0.223 (1.110)	−0.218 (−1.078)	−0.218 (−1.078)
rho1-rho2	−0.567 *** (−3.173)	−0.582 *** (−3.259)	−0.452 ** (−2.504)	−0.578 *** (−3.169)	0.469 ** (2.129)	0.493 ** (2.227)	0.491 ** (2.194)	0.491 ** (2.194)
Control Variables	YES	YES	YES	YES	YES	YES	YES	YES
Province FE	YES	YES	YES	YES	YES	YES	YES	YES
Time FE	YES	YES	YES	YES	YES	YES	YES	YES
R-squared	0.847	0.843	0.845	0.853	0.847	0.844	0.847	0.853
Obs	300	300	300	300	300	300	300	300

Note: T-value in parentheses, *, ** and *** indicate significant at the 10%, 5% and 1% statistical levels, respectively.

## Data Availability

The data presented in this study are available upon request from the corresponding author.

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
