# Peer review of "How Does Agricultural Mechanization Service Affect Agricultural Green Transformation in China?"

_ijerph, 2023, doi:10.3390/ijerph20021655_

Round 1

Reviewer 1 Report

Thank you for the opportunity to read the article. This paper measures the AGTFP using the Super-SBM model and examines the effects of different AMS supply agents on AGTFP and spatial spillover effects through the spatial Durbin model. I greatly enjoyed reading your article and I think that you are addressing a timely and relevant topic. However, after reading your research, I had some major concerns with your research, which prevented me from recommending its publication at the moment. I think that this article should be reconsidered by the editors after some major revisions. I will elaborate on my concerns below.

1.      Please revise and clarify the writing in the abstract. There are many instances of authors using the word ‘however’. Please simplify the abstract while increasing the readability of the main findings such as positive effect and opposite effect of some variables.

2.      Introduction is well-written. However, review of literature needs some revisions. Authors need to include review of existing methods of measuring AMS and its impacts (such as machine learning). In addition, more reviews of efficiency related studies is required. I am suggesting some relevant literature below:

https://doi.org/10.1016/j.scitotenv.2019.02.266

https://doi.org/10.1504/IJMTM.2021.121110

https://doi.org/10.3390/ijerph18031175.

3.      Please rename the heading 3.1 to “measurement of AGTFP”

4.      Table 3: The results are directly provided after the heading without any text description. Please avoid. Fit your results between the description text.

5.      Authors are expected to contextualize the headings in the results section.

6.      The general discussion merely presents an overview of the study findings, but it does not critically contextualize the study results in light of the extant scholarly debate. As a consequence, the authors are unable to provide us with thick and consistent information about how they are adding to the scholarly debate. Implications for theory and practice are limited. The authors do not provide adequate insights about how we can advance the scholarly knowledge about AMS and AGTFP. Besides, they do not provide us with compelling management implications supporting decision making processes in modern organizations.

7.      Please acknowledge the limitations of your research and consider expanding on future research directions.

8.      Format of the Tables, and Equations in the Materials and Methods section is not consistent.

9.      Please double-check all references, both in-text and in the bibliography. Check that the reference format is consistent with the journal style and that all references are current. You can remove some dated references while remaining true to the classics.

10.   Minor language improvements are required throughout the manuscript, so keep this in mind when submitting the revised version.

Reviewer 2 Report

1. The logical hierarchy and structural framework of the paper are clear, but some contents need to be optimized.

2. In the model building part of the article, if explanatory variables include core explanatory variables and control variables, it is suggested to accurately express them with summation symbols and give specific explanations.

3. In the time evolution characteristic plate of AGTFP, the content of text analysis is not clearly and intuitively presented in Table 3, so it is suggested to rearrange the presentation of the content in Table 3

4. The regression results in Table 4 are less in-depth analysis of the main effect and more statements of the regression results. It is suggested to analyze the relevant results to a certain extent instead of passing over them in one sentence. At the same time, what are the reasons for the inconsistency between the regression forms at the individual level and the organizational level? Can they be unified?

5. Policy inspiration is presented in the contribution part of the article, which is less and more scattered.

Round 2

Reviewer 1 Report

I have no further comments. The paper can be accepted for publication in its current form.

Author Response

We sincerely thank you for reviewing our manuscript entitled “How does agricultural mechanization service affect agricultural green transformation in China?” during your busy schedule. Thank you very much for your approval of our modifications. The quality of the manuscript cannot be improved without your valuable comments.

Reviewer 2 Report

1. In the part of model construction of the paper, the model expression has been modified, which meets the expectation of modification.

2. Table 3 adds relevant data of major and non-major grain-producing areas, economically developed areas and developing areas. The data matches the description analysis. In addition, if Table 3 is not the core part of the article, it is suggested to simplify this part of the text.

3. The main effect analysis of the regression results in Table 4 is added.

4. Policy suggestions have been modified to have a certain pertinency, and at the same time become relatively concentrated, not scattered as before.
